# Evaluation of Body Composition in CrossFit^®^ Athletes and the Relation with Their Results in Official Training

**DOI:** 10.3390/ijerph191711003

**Published:** 2022-09-02

**Authors:** Rubén Menargues-Ramírez, Isabel Sospedra, Francis Holway, José Antonio Hurtado-Sánchez, José Miguel Martínez-Sanz

**Affiliations:** 1Nursing Department, Faculty of Health Sciences, University of Alicante, San Vicente del Raspeig, 03690 Alicante, Spain; 2Research Group on Food and Nutrition (ALINUT), University of Alicante, San Vicente del Raspeig, 03690 Alicante, Spain; 3Faculty of Sport Science, Catholic University of Murcia (UCAM), Guadalupe de Maciascoque, 30107 Murcia, Spain

**Keywords:** CrossFit, cross-training, anthropometry, body composition, somatotype, athletic performance

## Abstract

CrossFit^®^ is a high-intensity sport that combines weightlifting, gymnastic skills, and cardiovascular exercises. To find the anthropometric references that define the optimal body composition, it is essential to first find an optimal body composition for one’s physical preparation. The objective of this study is to describe the anthropometric characteristics of 27 Spanish CrossFit^®^ athletes, 19 males aged 39 years old (24–44) and 8 females aged 28 years old (23–40), and how these characteristics influenced their performance. The athletes performed the Fran, Cindy, and Kelly workouts, establishing minimum marks, and the CrossFit Total workout to assess maximum strength. Significant differences were not found in time and repetitions between sexes in skill training, although there was a positive correlation r = 0.876 (*p* < 0.001) between muscle mass and the Total CrossFit result. We can conclude that the CrossFit^®^ athlete has a low amount of fat mass and a small relative size, which is an advantage when training with bodyweight exercises, and a high muscle mass that provide benefits when strength training. In addition, despite executing movements from a multitude of disciplines, the physical demands for lifting heavy loads resulted in the anthropometric values of athletes being more similar to elite weightlifting athletes than in other sports.

## 1. Introduction

CrossFit^®^ is a high-intensity sport created by Greg Glassman and Lauren Jenai [1,2] that combines weightlifting, typical of sports such as weightlifting and powerlifting, movements with body weight from artistic gymnastics, cardiovascular exercises, and various skills from other sports [3]. Since its creation, the presence of CrossFit^®^ has increased in the world of sports, with more than 13,000 official sports centers (boxes) around the world, 499 of them in Spain [4]. In addition, there are countless unofficial boxes where the same type of sports practice is carried out, commonly known as cross-training, but without the official license from the brand; so we can consider it as one of the sports that has grown the most in number of athletes in the last 20 years [5].

CrossFit^®^ workouts, called Workout of the Day (WOD), are extremely varied and can focus on one of the sports modalities that CrossFit^®^ is composed of or be a compendium of them. Thus, an athlete can perform both a WOD with their body weight that lasts no more than 5 min [6] and another workout that lasts 30–40 min that combines powerlifting, running, and movements on rings [7]. Therefore, this is a sport that not only builds an enormous physical capacity but also has a fairly high technical component from different sports modalities.

In competition, unlike in most sports, the athletes do not know which tests they are going to face until a few hours or minutes before it. Therefore, training, rest, feeding, and hydration in the days before and during the competition cannot be oriented towards specific parameters, as it is often found with athletes of more specific sports such as endurance [8]. Thus, the athletes must focus on maintaining an optimal general physical condition [9]. It should be noted that in CrossFit^®^, there is a single weight category, and all athletes have to perform the same exercises with the same loads regardless of their size, so it is not easy to find an ideal body composition (BC) as excess fat mass, or even muscle mass, could weigh you down in gymnastic movements, although they could be an advantage in heavy lifting of weightlifting [10].

Anthropometry is one of the tools used to estimate BC. It is a science that studies size, shape, proportionality, composition, biological maturation, and bodily function in order to understand the processes involved in growth, exercise, nutrition, and sports performance [11]. Today there is a standardized protocol created by the International Society for the Advancement of Kinanthropometry (ISAK) so that measurements are carried out with the highest possible validity and precision; which consists of taking anthropometric measurements: weight (kg), height (cm), sitting height (cm), arm span (cm), skinfolds (mm), girths (cm), lengths (cm) and bone diameters (mm); in order to assess people’s health status, body composition and somatotype [12].

Despite the growing interest of the sports population for CrossFit^®^, there are no studies that describe their athletes anthropometrically or that refer to BC or somatotype [13], and the suitability of one compared to another for the practice of this sport, as with other classic sports such as athletics [14], artistic gymnastics [15] or weightlifting [10,16].

It is considered that establishing anthropometric references that define the body composition of the athletes with the best marks is essential for the physical preparation of elite athletes [17]; however, we do not have that tool for CrossFit^®^.

The aim of this study is to describe the anthropometric characteristics, body composition, and somatotypes of CrossFit^®^ athletes from southeast Spain and how these parameters influence the performance of this sport. The main hypothesis of this research is that the studied sample will show differences according to sex and that CrossFit^®^ athletes will have a low amount of fat mass and a high muscle mass.

## 2. Materials and Methods

### 2.1. Design

An observational and descriptive pilot study of the anthropometric characteristics, body composition, and somatotype of CrossFit^®^ athletes from southeast Spain to find a correlation with their results in official WODs.

### 2.2. Sample

The study population was selected through non-probabilistic convenience sampling. The sample was made up of 19 men and 10 women, with an age range between 23 and 44 years, with high physical and technical levels, and an experience of >2 years in the practice of CrossFit^®^. In addition, the sample performs an average of 3 training per week. All athletes participated voluntarily, being informed of the entire process and methodology used, providing their consent in writing in accordance with the Declaration of Helsinki 2013 and the Ethics Committee from the University of Alicante (File UA-2020–03–29).

For the selection of athletes, we counted different official boxes that agreed to participate in the study. Athletes who met the inclusion criteria were recruited through the boxes and voluntarily accepted to participate.

The training sessions were carried out by the participating athletes at official CrossFit^®^ boxes, having the possibility of repeating some or all of the WODs, and only delivering the best mark for each of them at the end. Both the training and the anthropometric measurements were carried out in the months of February and March 2020.

### 2.3. Inclusion and Exclusion Criteria

For the selection of the sample, as CrossFit^®^ did not have official regional or national competitions with which to establish the minimum requirements of physical performance and experience in athletes, we decided to use official WODs, also known as Benchmark Workouts [18], which utilize standards defined by CrossFit^®^, both for technical execution and load (Rx). Training sessions were held on Mondays for four consecutive weeks as part of the participants’ pit schedule. This day was chosen since the schedules of these boxes consist of five pieces of training that take place from Monday to Friday, and in this way, it was guaranteed that the athlete carried out the training after having rested two days and not another day of the week when accumulated fatigue would interfere with their performance. For the selection of the official WODs, and considering the objective of this study, WODs chosen did not have high technical demand, and the physical abilities of the athletes were evaluated based on their body composition in addition to their skills or experience. The WODs were selected to cover the full range of CrossFit^®^ 100 workouts. These were: “Kipping Fran” [19], oriented towards strength/endurance, “Cindy” [19], which is performed with body weight, and “Kelly” [19], focused on aerobic endurance. In addition to these three WODs, which defined the suitability of the athletes for the study, another WOD, called “CrossFit Total” [19], was performed to assess maximum strength, in which the athletes had 15 min to perform three basic strength training exercises in one single repetition (1RM): Shoulder Press, Back Squat, and Dead Lift, recording the maximum weight lifted in each exercise and the total sum. We used the execution standards established by CrossFit^®^ through a CrossFit^®^ certified trainer and set the maximum execution times that would ensure an advanced physical level of the athletes in the sample. The data obtained were used to carry out the analysis (Table 1).

### 2.4. Materials and Procedures

To carry out this study, the following sociodemographic variables were collected for statistical or descriptive purposes, as well as for their use in the anthropometric formulas that required them: sex (male or female), age, ethnicity, and country of origin.

For taking the anthropometric values, the standards and measurement techniques recommended by the ISAK were used. The measurements were taken by an anthropometrist accredited in the ISAK level 2 who worked under the supervision of a level 3 accredited anthropometrist. We took into account the intraobserver technical error of measurement (ETM) indicated by the ISAK (5% for skinfolds and 1% for girths and diameters). Thus, the anthropometric measurements taken were those established by the ISAK [20] restricted profile: Basic (body mass, height, sitting height, arm span), Skinfolds (triceps, subscapularis, biceps, iliac crest, supraspinal, abdominal, thigh, leg), Girths (relaxed arm, flexed and contracted arm, waist, hips, middle thigh, leg) and Bone breadth (humerus, bistyloid, femur), in addition to the biacromial and biiliocrestal breadth, belonging to the complete profile of the ISAK, for a total of 23 anthropometric measurements.

For the calculation of BC, a four-component system was used through the application of the equations from Lee et al. [21] for muscle mass, Faulkner [22] for fat mass, Rocha [23] for bone mass. The residual mass was calculated as the remaining body mass minus the other three components, as recommended for use for adult athletes in the consensus document of the Spanish Federation of Sports Medicine [24]. These data were calculated both in kg and in percentages of total weight.

To calculate the somatotype, the three-component system (mesomorphy, endomorphy, and ectomorphy) proposed by Heath and Carter [13] was used, thereby establishing the mean somatotype and position in the somatochart of the high-level CrossFit^®^ athlete, for both men and women.

The proportionality of the athletes in the sample was assessed using the Phantom [25] methodology, which allows describing which anthropometric values stand out with respect to a phantom human being, a mixture of a metaphorical woman and man.

In addition, the sum of the values of the 8 skinfolds, corrected girth of the relaxed arm, corrected girth of the middle thigh, corrected girth of the leg, body mass index (kg/m^2^), waist-height index (waist girth/height), relative size and acromio-iliac index (biacromial breadth/biiliocrestal breadth) [12,17,26] were calculated.

The number of times, repetitions performed, and weights used in the WODs were recorded in a sheet, which indicated in detail what the athlete had to do in each training session. The exercises were supervised by trainers or judges, all certified by CrossFit^®^, to ensure that they were performed in compliance with the standards mentioned.

Variables such as diet control, sleeping time, hydration, or use of ergogenic sports supplements were not studied because the aim of the study was to describe anthropometric variables and how they can affect performance, but not to analyze what results were obtained depending on the lifestyles of the athletes.

The equipment used for the anthropometric measurements consisted of a Harpenden skinfold caliper (precision 0.2 mm), Cescorf anthorpometric tape (precision 1 mm), Cescorf segmometer (precision 1 mm), RealMet small sliding caliper (precision 1 mm), RealMet large sliding caliper (precision 1 mm), RealMet dermographic pen, Seca standing height rod (1 mm precision), Ozeri scale (100 g precision) and anthropometric drawer (40 × 50 × 30 cm); all of which were approved, correctly calibrated, and checked every six months.

### 2.5. Statistical Analysis

The statistical analysis was carried out through a descriptive and association study by sex of the anthropometric variables and the results of the training. Given the sample size, it was assumed that the values obtained did not have a normal distribution. Thus, we obtained the mean, standard deviations, median, minimum, and maximum for all values. Likewise, in order to find a relationship between the anthropometric values and the training results, the Spearman correlation was used. To find the statistical differences between sexes and the results of the WODs Fran, Cindy and Kelly, a Spearman correlation was calculated. All mathematical and statistical analyzes were performed with Microsoft Office Excel 2019 (Redmond, WA, USA) programs for MacOS, and IBM SPSS 24 (Armonk, NY, USA).

## 3. Results

Of the initial sample of 19 men and 10 women, 2 women were discarded because they did not meet the minimum physical level requirements (WODs) for participation in the study, finally leaving a sample of 19 men aged 39 (24–44) years old, and 8 women aged 28 years-old (23–40).

In relation to the results of the WODS, to find the statistical differences between sexes and the results of the WODs Fran, Cindy, and Kelly, a Spearman correlation was calculated, and it was observed that the significance between each of the WODs and sex was greater than 0.05 (*p* > 0.05), so we conclude that there are no differences between sexes. It was not the same with Crossfit Total, in which the significance of the correlation was *p* < 0.001. Table 2 shows that there were no significant differences between sexes in the training sessions that required technical skills (Kipping Fran r = 0.056, Kelly r = –0.297, Cindy r = 0.006). However, there was a significant correlation between the results in CrossFit Total and gender (r = –0.792); but since it is a maximum strength exercise, regardless of gender, this correlation is more related to muscle or bone mass, r = 0.876 (Figure 1) and r = 0.803, respectively.

We also found a correlation (*p* < 0.05) between the results that each athlete obtained in the Kipping Fran (r = 0.607), Cindy (r = –0.412), or Kelly (r = 0.572) WODs; however, there was no relationship between these three WODs and CrossFit Total. We must also highlight the relationship for men between the relative size and the results in the Kipping Fran (r = 0.643, *p* < 0.001).

Table 3 shows the anthropometric characteristics of the 27 CrossFit^®^ athletes in the sample, separated into men and women. The data were separated by sex and grouped into basic measurements. It was interesting to note that despite the participants being athletes with a level of physical fitness, the age range was between 23 and 44 years old. We also observed a wide range of measurements in other parameters, such as height, which ranged between 163.3 and 180.9 cm for men and between 160.3 and 169.7 cm for women.

The median value of the percentage of fat mass of the male population in our sample was 11.60%, and for the female population, it was 15.23%, which signifies a lean mass ranging between 60.8 and 79.3 kg in the case of men, and 45.1 and 57.1 kg in the case of women. In relation to body mass index (BMI), the median for men was 25.79 kg/m^2^ and 23.12 kg/m^2^ for women. Regarding somatotype results, CrossFit^®^ athletes showed endomorphic values of 2.0 (1.2–4.8) for men, and 2.6 (1.9–3.8) for women; and ectomorphic values of1.1 (0.1–2.6) and 1.86 (1.2–2.8) for men and women respectively.

Figure 2 shows the somatochart or graphic representation of the somatotype, where we can observe how both men and women had balanced endomorphy/ectomorphy proportions and, in both cases, a mesomorphy with values of 6.8 (5.1–8.6) for men, and 4.9 (3.2–5.2) for women.

Figure 3 shows the representation of the Z-Scores, where it was observed that the muscle mass of men was 2.56 standard deviations above the Phantom model, which places it above the 95% confidence interval (95% CI), such in the girth of the contracted arm; Likewise, most skinfolds, both in men and women, were found between one and two standard deviations below the general population, which places them below the 68% CI.

## 4. Discussion

Despite the growing number of CrossFit^®^ athletes, a complete anthropometric description of athletes in this discipline has not yet been established. Therefore, the present study aimed to establish a descriptive basis for this sport for future research.

The weight of elite Spanish athletes [17] is 75.2 ± 12.8 kg for men and 59.3 ± 9.9 kg for women; similar to those of our sample, 78.6 (66.5–96.3) (min–max) kg, and 61.80 (52.3–67.0) kg for men and women, respectively. This was also found for the athlete’s height, with an average of 179.5 ± 8.3 cm for men and 166.3 ± 7.4 cm for women, for Spanish athletes, and 173.6 (163.3–180.9) and 162.0 (160.3–169.7) for the men and women in our sample, respectively. Other studies have shown similar heights and weights for male CrossFit^®^ [27] athletes: 177.8 ± 7.3 cm and 83.8 ± 11.7 kg; and similar weights for women [28], 59.3 ± 5.7 kg. The bistyloid and humerus bone diameters above the 68% CI and 95% CI, respectively, are worth highlighting, which implies a high potential for muscle development in the upper extremities of the athletes [17].

The values obtained from the median BMI in the case of men were above 24.99 kg/m^2^, a figure that would indicate overweightness [29], and in some cases from our sample, it was even above 29.99 kg/m^2^, which would fall into the obesity range. In the case of women, the BMI values were between 20.33 and 24.37 kg/m^2^, which corresponded to the normal weight range in this index. However, BMI is not the most relevant index to take into account in athletes since it does not take into account which body compartment that mass refers to (fat, muscle, lean, or bone mass) [30].

Therefore, for the body composition analysis, the four-component model was utilized: muscle mass, fat mass, bone mass, and residual mass. According to this model, the results indicated that both men and women were found in fat mass percentage ranges that would be considered athletic [12,17].

Comparing these indices, it was found that 9 of the 19 men in the sample had a BMI that indicated overweightness or obesity, as compared to the percentage of fat in which the entire sample demonstrated healthy values. In other studies, in which BMI was compared with the percentage of fat, even greater differences were found, finding results of 97.7% overweight or obese according to the BMI, as compared to 8.9% according to the percentage of fat [31]; or another in which 72% of the athletes would have been misclassified as obese according to the BMI results [32].

The Acromio-iliac Index shows us the ratio between the width of the shoulders measured at the point of the acromion and the width of the hip at the height of the iliac crest. It is considered that there is a trapezoidal trunk when the proportion is less than 70% in men and 75% in women; intermediate when it is less than 75% in men and 80% in women; and rectangular in the rest of the cases. In the general population, these figures are 71.78% for men and 80.71% for women [33]. In our sample, the median for men was 69.04% (62.11–77.06), and for women, 70.15% (65.57–74.68). Of the men, 11 were found to be trapezoidal, 7 intermediates, and only 1 rectangular; and in the case of the women, all 8 were trapezoids. Although multiple studies [17,34,35,36,37] have concluded that the most common trunk type in the sports population is the trapezoidal trunk, no studies have been found that relate this index to sports performance. Some works relate limb length to weightlifting power and velocity, but a larger skeletal structure (frame) would be advantageous for the accumulation of muscle mass [38,39].

We compared our sample with weightlifting and artistic gymnastics athletes according to the study by Victoria Pons et al. [40], and we observed that values such as body mass (79.3 ± 8.3 kg in our sample, compared to 76.5 ± 13.1 kg in the sample from the study mentioned), height (174.0 ± 4.5 cm, versus 172.1 ± 6.3 cm), fat percentage (12.17 ± 2.80% versus 13.60 ± 4.80%), or the somatotype components (X = −1.1 ± 1.7 vs. −1.7 ± 2.5 and Y = 9.7 ± 2.5 vs. 7.2 ± 3.0), were similar between our sample and weightlifting athletes, but we did not find any similarities with the artistic gymnastics athlete sample. This fact is striking, as CrossFit^®^ is composed of a multitude of gymnastic exercises. However, perhaps the physical demands when lifting heavy loads will result in the anthropometric values of CrossFit^®^ athletes becoming more similar to those of weightlifting athletes, with the exercises from artistic gymnastics performed in a more basic manner and with execution based more on strength than technique.

If the anthropometric parameters of the studied sample stand out in something, it is in the heterogeneity of their values, which makes sense given the variability of exercises practiced, each with completely different physical demands that gave advantages to some athletes over others but which makes it extremely difficult to find that perfect anthropometric profile for the practice of this particular sport.

Although our results showed differences by sex in CrossFit Total when analyzing results from the WODs that required technical skills (Kipping Fran, Kelly, and Cindy), we did not find significant differences between sexes. These WODs are adapted for men and women both in the loads lifted (for women, 2/3 of the loads lifted by men) and in the execution of some exercises to try to balance the results between sexes; therefore, the adaptations made in the CrossFit [9] regulations seems to be correct, due to the fact that avoid the differences caused by sex.

However, a correlation (r = 0.643, *p* < 0.001) was found between a short arm span in men and better results in Kipping Fran, in which the speed of execution of the pull-ups is decisive for the completion time of this workout; which coincides with the results found in a study by Taboada-Iglesias et al. [36], in which they concluded that short upper limbs were advantageous in artistic gymnastics.

In a study by Butcher et al. [19], as in our study, they faced a group of athletes against the CrossFit Total and Cindy WODs. The participants in their sample had competed nationally and some internationally in 2014. These athletes achieved a result of 401.5 ± 83.1 kg on the CrossFit Total and 698 ± 113 reps on the Cindy WODs. Our male athletes achieved results of 355 (305–420) kg and 555 (470–801) reps, respectively. Somewhat lower results, but within the same ranges. We can conclude, therefore, that the inclusion criteria of our sample were correct for the selection of a sample with a high level of physical fitness, without the need to limit ourselves exclusively to those who competed professionally in this discipline, thereby indicating that the same criteria could be used to select a larger sample in future studies.

The limitations that we have found in the present study were: the small geographical area given the time established for the study and its nature as a pilot study. Another limitation is the sample size of the athletes recruited, and this could influence the statistical analysis carried out. Anyway, small sample sizes characterize the studies carried out so far on this subject in strength sports. Furthermore, the selection of the sample became complex as there were no regional or national competitions where we could find the athletes, and for this reason, we used official WODs that any experienced CrossFit^®^ athlete would have performed one or several times in recent years. Despite not being an objective of this study, if variables such as diet, sleep, hydration, or use of ergogenic aids had been controlled, the relationship between anthropometric variables and sports performance could have been better isolated. Thus, not knowing the training demands or how to manage energy during its execution were no factors that limited the performance of the athletes in each test. Despite all of this, and the limitation of the sample obtained, it is the only study that can be currently found in the scientific literature that provides a complete anthropometric description of CrossFit^®^ athletes and that offers data on how these data influence sports performance. Future research should address how and why anthropometric characteristics and body composition might influence CrossFit performance.

## 5. Conclusions

In view of the results, we can conclude that the CrossFit^®^ athlete has a low amount of fat mass and a small relative size, which is an advantage in the WODs, which include exercises performed with body weight, and a high muscle mass that provides benefits in strength training. An acromio-iliac index under 70% in men and around 70% in women seems to be a desirable characteristic for CrossFit^®^ athletes, although there is a lack of evidence about the influence of the AI index on performance.

Given the heterogeneity of the anthropometric data of the CrossFit^®^ athletes, it is difficult to compare the results obtained with other studies in similar sports.

## Figures and Tables

**Figure 1 ijerph-19-11003-f001:**
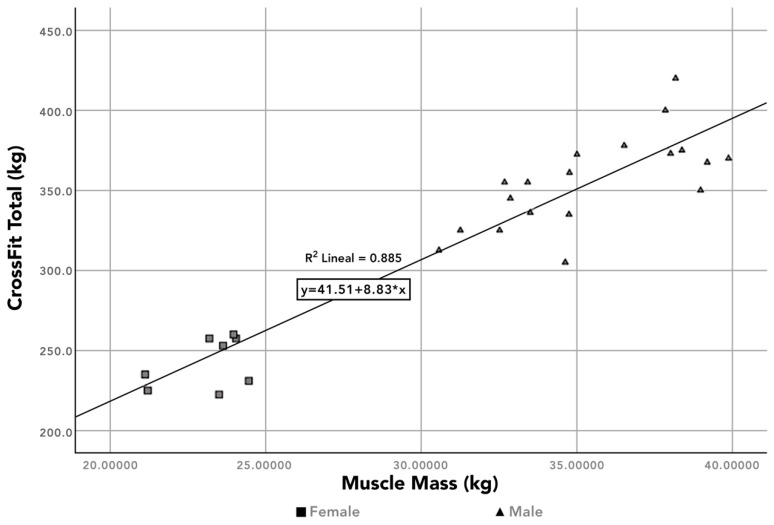
Correlation between muscle mass and weight lifted in CrossFit Total.

**Figure 2 ijerph-19-11003-f002:**
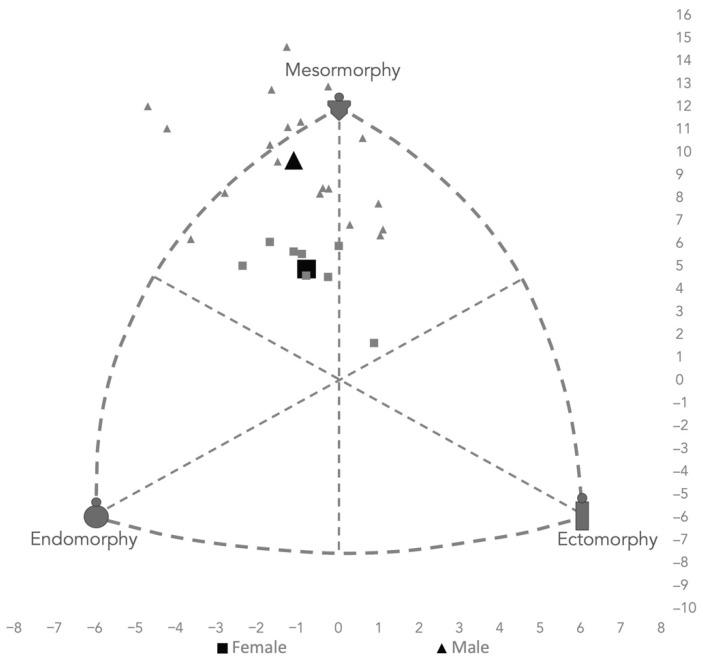
Somatochart of the entire sample and mean values. Note: The small shapes represent the individual values of the sample, and the large ones, the mean values.

**Figure 3 ijerph-19-11003-f003:**
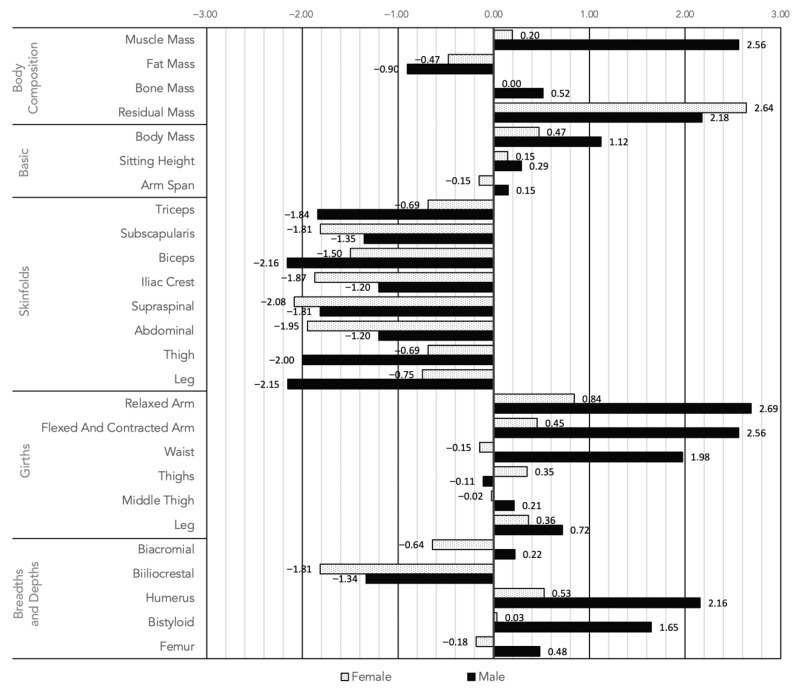
Phantom of the sample for men and women.

**Table 1 ijerph-19-11003-t001:** WODs requirement for sample selection.

**(Kipping ^1^) Fran**	**Kelly**
** *21-15-9 Repetitions per Time* **	** *5 Rounds for Time* **
- Thrusters (95/65 lbs.) - Pull-Ups	- 400 m Run - 30 Box Jumps (24/20 inch) - 30 Wall Ball Shots (20/14 lbs.)
**Time Cap ^2^: 9′**	**Time Cap: 30′**
**Cindy**	**CrossFit Total**
** *AMRAP* ^3^ *in 20′* **	** *Sum of the Best of Each Lift* **
- 5 Pull-Ups - 10 Push-Ups - 15 Air Squats	- 1RM ^4^ Back Squat - 1RM Shoulder Press - 1RM Dead Lift
**Minimum 12 sets**	**Time Cap: 15′**

^1^. Kipping: Hanging pull-up exercise with momentum. ^2^. Time Cap: time limit. ^3^. AMRAP: As Many Reps As Possible. ^4^. 1RM: 1 repetition maximum.

**Table 2 ijerph-19-11003-t002:** Results of the WODs for the selection of the sample.

	Male				Female			
	n	Mean ± Std Dev.	Median	Min–Max	n	Mean ± Std Dev.	Median	Min–Max
Kipping Fran (s)	18	372 ± 78	349	233–523	6	375 ± 534	393	292–430
Kelly (s)	18	1806 ± 174	1795	1373–2082	7	1712 ± 180	1660	1472–2027
Cindy (reps)	19	577 ± 85	555	470–801	7	566 ± 85	605	453–660
Shoulder Press (kg)	19	63.8 ± 7.3	63,0	52.5–80.0	8	43.3 ± 4.2	42,8	37.5–50.0
Back Squat (kg)	19	126.2 ± 11.4	125.0	110.0–160.0	8	89.8 ± 5.5	90.0	83.0–100.0
Dead Lift (kg)	19	165.8 ± 16.6	170.0	140.0–190.0	8	109.7 ± 10.4	110.0	97.5–125.0
CrossFit Total (kg)	19	355.8 ± 29.1	355.0	305.0–420.0	8	242.7 ± 15.9	244.0	222.5–260.0

**Table 3 ijerph-19-11003-t003:** Anthropometric values of the sample of CrossFit athletes.

		Male (n = 19)			Female (n = 8)		
		Mean ± Std Dev	Median	Min–Max	Mean ± Std Dev.	Median	Min–Max
Basics	Age (years)	37 ± 6	39	24–44	30 ± 7	28	23–40
	Body mass (kg)	79.3 ± 8.3	78.6	66.5–96.3	60.9 ± 5.2	61.8	52.3–67.0
	Stature (cm)	174.0 ± 4.5	173.6	163.3–180.9	163.5 ± 3.4	162.0	160.3–169.7
	Sitting height (cm)	93.3 ± 2.2	93.5	89.4–97.3	87.0 ± 1.6	86.9	85.4–90.2
	Arm span (cm)	177.4 ± 6.1	178.0	160–185.2	164.5 ± 4.6	166.1	155.2–168.5
Skinfolds (mm)	Triceps	7.3 ± 2.9	6.5	4.0–13.8	11.8 ± 2.8	12.3	8.0–16.6
	Subscapular	10.6 ± 4.7	9.1	6.1–21.7	7.7 ± 1.7	7.1	5.9–11.1
	Biceps	3.8 ± 1.3	3.5	2.2–7.0	4.8 ± 1.8	3.8	3.0–7.7
	Iliac crest	14.5 ± 8.3	12.5	5.3–34.2	9.3 ± 1.5	9.3	7.2–11.2
	Supraspinale	7.5 ± 3.7	6.2	3.6–16.6	5.8 ± 0.9	5.7	5.0–7.8
	Abdominal	16.4 ± 8.4	16.0	4.9–34.1	9.8 ± 0.9	9.8	8.7–11.4
	Thigh	10.6 ± 4.2	10.2	5.0–22.0	20.5 ± 4.3	20.7	13.5–27.1
	Calf	6.1 ± 2.7	5.3	2.8–12.6	12.0 ± 3.7	12.0	7.3–18.4
	∑ 8 skinfolds	76.7 ± 32.3	67.6	37.0–136.7	81.8 ± 13.4	79.2	64.6–103.3
Girths (cm)	Arm (relaxed)	33.9 ± 2.2	33.5	29.7–37.5	27.7 ± 1.4	27.9	25.2–30.0
	Arm (flexed and tensed)	36.3 ± 1.9	36.1	33.5–39.9	29.3 ± 139	29.4	26.7–31.0
	Waist (minimum)	82.5 ± 6.1	81.5	74.2–98.0	68.5 ± 3.6	69.4	62.8–73.5
	Hip (maximum)	96.1 ± 5.3	96.2	88.2–106.0	92.8 ± 3.8	93.5	87.9–98.3
	Thigh middle	55.4 ± 3.4	55.1	50.7–61.9	51.0 ± 2.5	51.5	46.3–53.8
	Calf (maximum)	37.7 ± 2.4	37.5	33.5–42.2	34.7 ± 1.8	34.6	31.5–36.9
Breadths (cm)	Biacromial	39.3 ± 1.5	39.4	37.7–43.4	35.3 ± 1.6	35.7	33.3–37.8
	Biiliocristal	27.1 ± 1.5	27.0	25.0–29.6	24.7 ± 1.3	24.9	22.4–26.4
	Humerus	7.4 ± 0.3	7.4	6.6–8.2	6.4 ± 0.3	6.5	5.7–6.6
	Bi-styloid	5.8 ± 0.3	5.8	5.3–6.3	5.0 ± 0.3	5.0	4.7–5.5
	Femur	10.0 ± 0.4	9.9	9.4–10.8	9.1 ± 0.4	9.0	8.4–9.6
Body composition (kg)	Lean mass	69.46 ± 5.59	68.93	60.67–79.31	51.52 ± 4.33	51.50	45.05–57.10
	Muscle mass	35.42 ± 2.88	34.76	30.57–39.88	23.14 ± 1.28	23.57	21.13–24.46
	Fat mass	9.82 ± 3.22	8.71	5.78–16.99	9.39 ± 1.15	9.89	7.25–10.93
	Bone mass	12.06 ± 0.73	12.10	10.88–13.11	9.31 ± 0.77	9.23	8.19–10.77
	Residual mass	21.98 ± 3.23	22.35	16.83–28.52	19.07 ± 2.83	18.64	15.73–23.67
	% Muscle mass	44.86 ± 3.02	44.68	38.83–50.38	38.11 ± 2.03	37.88	35.78–41.60
	% Fat mass	12.17 ± 2.80	11.60	8.71–17.64	15.40 ± 1.25	15.23	13.86–17.89
	% Bone mass	15.31 ± 1.22	15.54	12.27–16.67	15.31 ± 0.81	15.56	13.99–16.40
	% Residual mass	27.66 ± 2.08	28.03	22.17–30.29	31.18 ± 2.24	30.38	28.66–35.33
Anthropometric indexes	Body mass index (kg/m^2^)	26.19 ± 2.59	25.79	23.31–32.23	22.77 ± 1.34	23.12	20.33–24.37
	Waist-Stature Index	0.47 ± 0.04	0.46	0.43–0.57	0.42 ± 0.02	0.42	0.39–0.44
	Relative Span	101.94 ± 1.86	102.02	98.01–104.84	100.61 ± 2.12	101.22	96.73–103.29
	Acromio-Iliac Index	68.96 ± 3.99	69.04	62.11–77.06	69.81 ± 3.51	70.15	65.57–74.68
Somatotype	Endomorphy	2.4 ± 1.1	2.0	1.2–4.8	2.7 ± 0.6	2.6	1.9–3.8
	Mesomorphy	6.7 ± 1.1	6.8	5.1–8.6	4.7 ± 0.6	4.9	3.2–5.2
	Ectomorphy	1.3 ± 0.7	1.1	0.1–2.6	1.9 ± 0.6	1.9	1.2–2.8

## Data Availability

The data presented in this study are available in the tables of this article. The data presented in this study are available on request from the corresponding author.

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
