# Peer review of "Evaluation of Body Composition in CrossFit® Athletes and the Relation with Their Results in Official Training"

_ijerph, 2022, doi:10.3390/ijerph191711003_

Round 1

Reviewer 1 Report

The present investigation seeks to investigate the role of body composition (i.e., lean mass) on CrossFit performance in CrossFit-trained males and females from Spain. The reviewer wishes to help improve the science and hopes the comments are helpful to the authors for future endeavors. This topic is extremely relevant given that limited data exist regarding CrossFit, and that this type of exercise is increasingly popular. Two main areas need to be addressed to strengthen the manuscript. First, please review the statistical analysis, especially regarding the sex differences and sample sizes as the methodology does not match the interpretations in the discussion. Second, please bolster the discussion by improving the interpretation of the results and ensure all results are accounted for. This will clarify why and how these results will inform CrossFit training in the future. The reviewer sincerely hopes the comments are helpful to the scientists for future endeavors.

Introduction

-       Please add a hypothesis/hypotheses.

Methods

-       How was sample size calculated? Was there a power calculation based on reports from other body composition studies in other exercise modalities (e.g., weightlifting, gymnastics)?

-       Please add the requirements for determining CrossFit experience – line 86 mentions at least 2 years minimum but what about current number of times the individuals perform CrossFit? For example, 2-3 x per week?

-       Please add specific age range of subjects and replace ‘legal age’

-       Please add whether there were controls for diet, sleep, hydration, and other lifestyle factors (e.g., dietary and/or ergogenic supplements) or provide a statement for the reason if not.

-       Please add whether statistical analysis was conducted to analyze for sex-differences, or provide indication of why it was not conducted.

Results

-       Line 180/181 please superscript the 2 for m2 – this is also an issue in the discussion so please amend throughout the whole document.

-       Table 2 – the headings for ‘anthropometric’ and ‘body composition’ are cut off – please fix.

-       Line 199 to 201 fits more in the discussion section. Please relocate to the discussion as this section should simply state results without interpretation.

-       Figure 1 – please clarify in the figure description what the smaller vs. larger shapes mean.

-       Table 3 – would it be more clear to make this Table 1 and display it earlier in the results section? That way, it would match the order that the variables were presented in the methods section.

-       Line 205 – please clarify in the methods statistics section what statistical methods were used to analyze for sex

Discussion

-       Line 237 – citation 32 missing brackets.

-       Line 247-256 – please include statements to relate these findings to how it is relevant to performance. How are the AI-index results helpful in determining performance? What does this mean in terms of training for CrossFit individuals?

-       Line 274 – how did the authors analyze sex differences, especially when the sample size is smaller for females than the males? Also, the males have sample sizes of 18 and 19 depending on the variable so please clarify the methodologies and then update this interpretation.

-       Line 288-289 – regarding the phrasing ‘statistically different’ – this implies the authors retrieved raw data from this other publication and conducted statistical analyses but it is not outlined in the methods section. If this was the case, please include statistical analyses for this. Or if the authors did not obtain data from this study to run the statistical difference, remove/update sentence to not contain the word ‘statistically’

-       Line 293 – remove extra period.

-       Please add discussion on the limitations regarding the lack of control for lifestyle factors and supplementation, or add the control information to the methods.

-       There was no discussion of the skinfold results. If the variables are presented in the results, there should be an interpretation relative to the hypothesis (which is also missing). A suggestion is to add the skinfold interpretation alongside the BMI information to further reinforce the body mass story of the findings.

-       Consider including discussion on anthropometric data in power-based sports rather than weightlifting and gymnastics since CrossFit may utilize similar energetic demands in power-based sports such as rugby.

-       Conclusions – add interpretation and findings regarding the AI index. Add significance of this data – how will the CrossFit community utilize this data? What are future areas of research? Why is this work important? How can this data inform CrossFit training?

Author Response

Author's Reply to the Reviewer 1 - ijerph- 1858421 "Evaluation of body composition in CrossFit® athletes and the relation with their results in official training”.

Reviewer 1

The present investigation seeks to investigate the role of body composition (i.e., lean mass) on CrossFit performance in CrossFit-trained males and females from Spain. The reviewer wishes to help improve the science and hopes the comments are helpful to the authors for future endeavors. This topic is extremely relevant given that limited data exist regarding CrossFit, and that this type of exercise is increasingly popular. Two main areas need to be addressed to strengthen the manuscript. First, please review the statistical analysis, especially regarding the sex differences and sample sizes as the methodology does not match the interpretations in the discussion. Second, please bolster the discussion by improving the interpretation of the results and ensure all results are accounted for. This will clarify why and how these results will inform CrossFit training in the future. The reviewer sincerely hopes the comments are helpful to the scientists for future endeavors.

Response of the authors: According to reviewer comments, the manuscript has been revised and improved taking into account reviewer’s comments.

 Introduction

-       Please add a hypothesis/hypotheses.

 Response of the authors: According to reviewer comment, a hypothesis has been added at the end of the introduction section.

Methods

-       How was sample size calculated? Was there a power calculation based on reports from other body composition studies in other exercise modalities (e.g., weightlifting, gymnastics)?

Response of the authors: The study population was selected through non-probabilistic convenience sampling and all the athletes who met the inclusion criteria were included in the study. This information has been included in sample section to clarify the manuscript.

-       Please add the requirements for determining CrossFit experience – line 86 mentions at least 2 years minimum but what about current number of times the individuals perform CrossFit? For example, 2-3 x per week?

Response of the authors: Athletes included had at least 2 years of experience and currently they were training at least 3 times a week. These data have been added in material and methods section.

-       Please add specific age range of subjects and replace ‘legal age’

Response of the authors: According to reviewer comments, the age range of the sample has been added and the term ‘legal age’ has been replaced.

-       Please add whether there were controls for diet, sleep, hydration, and other lifestyle factors (e.g., dietary and/or ergogenic supplements) or provide a statement for the reason if not.

Response of the authors: Authors appreciate the reviewer's comments but these factors were not taken into account because they were not related to the objective of the study, but this aspect has been mentioned in the material and methods and limitations section.

-       Please add whether statistical analysis was conducted to analyze for sex-differences, or provide indication of why it was not conducted.

Response of the authors: The analysis by sex has been conducted and the results have been better explained in results section.

Results

-       Line 180/181 please superscript the 2 for m2 – this is also an issue in the discussion so please amend throughout the whole document.

Response of the authors: In accordance with the reviewer's suggestions, the modification has been made throughout the manuscript.

-       Table 2 – the headings for ‘anthropometric’ and ‘body composition’ are cut off – please fix.

Response of the authors: the authors have corrected this mistake.

-       Line 199 to 201 fits more in the discussion section. Please relocate to the discussion as this section should simply state results without interpretation.

Response of the authors: this sentence has been changed to the discussion section (second paragraph).

-       Figure 1 – please clarify in the figure description what the smaller vs. larger shapes mean.

Response of the authors: According to reviewer´s comments, figure 1 has been clarified.

-       Table 3 – would it be more clear to make this Table 1 and display it earlier in the results section? That way, it would match the order that the variables were presented in the methods section.

Response of the authors: According to reviewer´s comments, the order of presentation of the results in the results section has been modified. Table 3 becomes Table 2 and Figure 3 becomes Figure 1.

-       Line 205 – please clarify in the methods statistics section what statistical methods were used to analyze for sex

Response of the authors: According to reviewer’s comments, the information about the statistical methods used to analyze sex differences has been added into material and methods section.

 Discussion

-       Line 237 – citation 32 missing brackets.

Response of the authors: the mistake has been resolved.

-       Line 247-256 – please include statements to relate these findings to how it is relevant to performance. How are the AI-index results helpful in determining performance? What does this mean in terms of training for CrossFit individuals?

Response of the authors: Although several studies have described the most typical AI-index values in the sports population, no studies about the relation of this index with performance parameters have been found. The authors have described this situation to clarify the text.

-       Line 274 – how did the authors analyze sex differences, especially when the sample size is smaller for females than the males? Also, the males have sample sizes of 18 and 19 depending on the variable so please clarify the methodologies and then update this interpretation.

Response of the authors: the information about the analysis by sex has been added in material and methods section, and the corresponding discussion paragraph has been rewritten to clarify the interpretation of the results.

-       Line 288-289 – regarding the phrasing ‘statistically different’ – this implies the authors retrieved raw data from this other publication and conducted statistical analyses but it is not outlined in the methods section. If this was the case, please include statistical analyses for this. Or if the authors did not obtain data from this study to run the statistical difference, remove/update sentence to not contain the word ‘statistically’

Response of the authors: according to reviewer comment and having into account that the authors did not obtain data from this cited study, the word ‘statistically’ has been deleted.

-       Line 293 – remove extra period.

Response of the authors: the extra period has been removed.

-       Please add discussion on the limitations regarding the lack of control for lifestyle factors and supplementation, or add the control information to the methods.

Response of the authors: According to reviewer´s comments, the limitations section has been modified and improved to include these aspects.

-       There was no discussion of the skinfold results. If the variables are presented in the results, there should be an interpretation relative to the hypothesis (which is also missing). A suggestion is to add the skinfold interpretation alongside the BMI information to further reinforce the body mass story of the findings.

Response of the authors: Skin folds were considered as primary data that was used to estimate the fat mass and endomorphic component of the somatotype, aspects that have been widely discussed. Although skin folds can be studied independently, these data can greatly lengthen the manuscript without providing relevant information.  For this reason, the authors consider of greater interest to include only the discussion of the parameters obtained from the analysis of several of the skin folds.

-       Consider including discussion on anthropometric data in power-based sports rather than weightlifting and gymnastics since CrossFit may utilize similar energetic demands in power-based sports such as rugby.

Response of the authors: Authors appreciate reviewer suggestion, however, in our opinion, the comparison with athletes of weightlifting or gymnastics could be more relevant because they are cyclic sports in which the same pattern of movement is repeated and therefore proportionality and body composition affects much more than in acyclic sports (such as rugby or football) in which the movement pattern is different in each play and therefore the body composition and proportionality, although important, is in the background, and the strategy or experience technique are more important. In addition to that, in rugby, depending on the player position each one will have a completely different body composition, being heavier players if they do a defense job and lighter ones if they are runners. Also times are different, because CrossFit® WODs last between 4 and 30 minutes and a rugby match 80.

-       Conclusions – add interpretation and findings regarding the AI index. Add significance of this data – how will the CrossFit community utilize this data? What are future areas of research? Why is this work important? How can this data inform CrossFit training?

Response of the authors: Information about AI index has been added to conclusions section.

Although there is a lack of evidence about the influence of AI index on performance, our results show that a trapezoidal trunk with an AI Index under 70% in men and around 70% in women seems to be a common characteristic for the best CrossFit® athletes, so it should be a factor to take into account. To carry out new studies that relate this index to the specific performance of CrossFit® athletes can be decisive for detecting talents in this sport.

Reviewer 2 Report

Menargues Ramírez_Body composition_IJERPH_2022 

I commend the authors on the completion of this manuscript. 

The article includes a comprehensive introduction and background. 

The research question is well defined, being clinically relevant. The presentation defines the research question. The research has been carried out in accordance with ethics international recommendations, but I have some relevant concerns highlighted below.

General comment

Please, review the entire Material and Methods section and: 

- my recommendation is to divide the section in subsections, explaining independently the data collection techniques of the variables and the study protocol. 

- Variables must be better defined, including also in this section the measure magnitude, for individual measures as skinfolds or for index as the three-component of the somatotype. 

- clarify when the data about official WODs were collected. Was it in an unspecific training day or in a special event? The WODs data used for inclusion of the participants were the same as the used for the analysis and outcomes? What do you understand with maximum execution times that would ensure an advanced physical level of the athletes in the sample?

- Avoid the inclusion of variables in the material and methods section whose outcomes are not presented in the results section. 

Specific comments

Introduction

Line 45. “Therefore, this is a sport in which not only is an enormous physical capacity necessary,”, please, change to: Therefore, this is a sport in which not only is necessary an enormous physical capacity,”

Materials and Methods

Line 78: “We carried out an observational and descriptive pilot study of the anthropometric”. I think it is better: “a descriptive and analytical cross-sectional pilot study…”

Line 97: “For the selection of these official WODs, and considering the objective of  this study, WODs were chosen that did not have a high technical demand, and that 

evaluated the physical abilities of the athletes based on their body composition over their  skills or experience. The WODs were selected to cover the full range of CrossFit® 1workouts.” Please correct to: “For the selection of these official WODs, and considering the objective of this study, WODs chosen did not have a high technical demand, and evaluated the physical abilities of the athletes based on their body composition in addition to their skills or experience. The WODs were selected to cover the full range of CrossFit® 100 workouts.”

Line 103: “We used the execution standards established by CrossFit®, and set the maxi- mum execution times that would ensure an advanced physical level of the athletes in the  sample”. Please, clarify in what way maximum execution times were stablished. 

Line 129: “For the calculation of WC, a four-component system…”. Please correct to: “For the calculation of BC, a four-component system…”

Line 135: “To calculate the somatotype, the three-component system (mesomorphy, endomor phy, and ectomorphy) proposed by Heath and Carter [13] was used, thereby establishing  the mean somatotype and position in the somatochart of the high-level CrossFit® athlete, for both men and women.” What values can each component of the somatype take? What value can they take in total?

Line 142: “In addition, the sum of the values of the 8 skinfolds, corrected girth of the relaxed 142 arm, corrected girth of the middle thigh, corrected girth of the leg, fat free mass index, 143 body mass index, waist-hip index, waist-height index, adipose-muscular index, muscle-144 bone index, fat distribution index, relative size and acromio-iliac index [12,17,26] were 145 calculated.” Please include the formulas for corrected values and index. 

Results

Line 181: “As for their somatotype, CrossFit® athletes showed an endomorphy and ectomorphy of 2.0 [1.2-4.8] for men, and 2.6 [1.9-3.8] for women; and 1.1 [0.1-2.6] and 1.86 [1.2-2.8], respectively.” Please, review English editing.

Line 208: “but since it is a maximum strength exercise, regardless of gender, this correlation is more related to mus-209 cle or bone mass, r = 0.876 (figure 3) and r = 0.803, respectively.” Where is figure 3?

Discussion

Line 225: “173.6  [163.3-180.9] and 162.0 [160.3-169.7]” Please, identify the values as [min-max], if not they may be confounded with CI. 

Line 288: “Somewhat lower results, but statistically within the same ranges.” I think this sentence need a more concrete justification. How do you know they are in the same statistically ranges?

Author Response

Author's Reply to the Reviewer 2 - ijerph- 1858421 "Evaluation of body composition in CrossFit® athletes and the relation with their results in official training”.

Reviewer 2

I commend the authors on the completion of this manuscript. 

The article includes a comprehensive introduction and background. 

The research question is well defined, being clinically relevant. The presentation defines the research question. The research has been carried out in accordance with ethics international recommendations, but I have some relevant concerns highlighted below.

General comment

Please, review the entire Material and Methods section and: 

- my recommendation is to divide the section in subsections, explaining independently the data collection techniques of the variables and the study protocol. 

Response of the authors: According to reviewer´s comments, the material and methods section has been divided into subsections.

- Variables must be better defined, including also in this section the measure magnitude, for individual measures as skinfolds or for index as the three-component of the somatotype. 

Response of the authors: Authors appreciate reviewer suggestion, and although this information is relevant, a more specific description would greatly lengthen the manuscript by providing information not necessary for the target audience because of its expertise. So, authors consider that, as has been done in previously published works (https://doi.org/10.3390/ijerph18020756), the current description fits the needs of the journal.

- clarify when the data about official WODs were collected. Was it in an unspecific training day or in a special event? The WODs data used for inclusion of the participants were the same as the used for the analysis and outcomes? What do you understand with maximum execution times that would ensure an advanced physical level of the athletes in the sample?

Response of the authors: According to reviewer´s comments, this information has been clarified in material and methods section.

- Avoid the inclusion of variables in the material and methods section whose outcomes are not presented in the results section. 

 Response of the authors: According to reviewer´s comments, variables not described in results have been removed, only those primary variables that have been used to calculate other variables or indexes have been retained in the text.

Specific comments

Introduction

Line 45. “Therefore, this is a sport in which not only is an enormous physical capacity necessary,”, please, change to: “Therefore, this is a sport in which not only is necessary an enormous physical capacity,”

 Response of the authors: the sentence has been modified

Materials and Methods

Line 78: “We carried out an observational and descriptive pilot study of the anthropometric”. I think it is better: “a descriptive and analytical cross-sectional pilot study…”

 Response of the authors: the sentence has been modified

Line 97: “For the selection of these official WODs, and considering the objective of this study, WODs were chosen that did not have a high technical demand, and that evaluated the physical abilities of the athletes based on their body composition over their skills or experience. The WODs were selected to cover the full range of CrossFit® 1workouts.” Please correct to: “For the selection of these official WODs, and considering the objective of this study, WODs chosen did not have a high technical demand, and evaluated the physical abilities of the athletes based on their body composition in addition to their skills or experience. The WODs were selected to cover the full range of CrossFit® 100 workouts.”

Response of the authors: the sentence has been modified

Line 103: “We used the execution standards established by CrossFit®, and set the maxi- mum execution times that would ensure an advanced physical level of the athletes in the sample”. Please, clarify in what way maximum execution times were stablished. 

Response of the authors: The sentence has been modified to clarify the meaning.

Line 129: “For the calculation of WC, a four-component system…”. Please correct to: “For the calculation of BC, a four-component system…”

Response of the authors: the sentence has been modified.

Line 135: “To calculate the somatotype, the three-component system (mesomorphy, endomor phy, and ectomorphy) proposed by Heath and Carter [13] was used, thereby establishing the mean somatotype and position in the somatochart of the high-level CrossFit® athlete, for both men and women.” What values can each component of the somatype take? What value can they take in total?

Response of the authors: The values are between 1 and 8.5 but can range as follows:

  • Endomorphy: 1/2-16,
  • Mesomorphy: 1/2-12
  • Ectomorphy: 1/2-10

This information is available through the citation indicated by the authors, but can also be found at https://www.researchgate.net/publication/283664365_Antropometrica_Spanish_version_of_Anthropometrica_Norton_K_and_T_Olds_1995

This information can be of interest for the readers and is accessible through the citation included in the text. However, and since somatotype calculation is a widely used technique, the authors consider that its inclusion in the manuscript would only lengthen the text unnecessarily.

Line 142: “In addition, the sum of the values of the 8 skinfolds, corrected girth of the relaxed 142 arm, corrected girth of the middle thigh, corrected girth of the leg, fat free mass index, 143 body mass index, waist-hip index, waist-height index, adipose-muscular index, muscle-144 bone index, fat distribution index, relative size and acromio-iliac index [12,17,26] were 145 calculated.” Please include the formulas for corrected values and index. 

Response of the authors: formulas have been entered.

Results

 Line 181: “As for their somatotype, CrossFit® athletes showed an endomorphy and ectomorphy of 2.0 [1.2-4.8] for men, and 2.6 [1.9-3.8] for women; and 1.1 [0.1-2.6] and 1.86 [1.2-2.8], respectively.” Please, review English editing.

Response of the authors: the sentence has been modified to clarify the manuscript.

Line 208: “but since it is a maximum strength exercise, regardless of gender, this correlation is more related to mus-209 cle or bone mass, r = 0.876 (figure 3) and r = 0.803, respectively.” Where is figure 3?

Response of the authors: the authors have introduced figure 3 (now figure 1).

Discussion

Line 225: “173.6 [163.3-180.9] and 162.0 [160.3-169.7]” Please, identify the values as [min-max], if not they may be confounded with CI. 

Response of the authors: The manuscript has been modified to clarify the meaning according to reviewer’s suggestion.

Line 288: “Somewhat lower results, but statistically within the same ranges.” I think this sentence need a more concrete justification. How do you know they are in the same statistically ranges?

Response of the authors: the sentence has been modified to be more clarifying.

Round 2

Reviewer 1 Report

Thank you for addressing the comments. Future studies related to this topic should include more interpretation on the correlation results to offer more insight into how and why body composition might influence crossfit performance.

Author Response

We appreciate reviewer's comments and hope that this research lays the groundwork for future research in CrossFit. Therefore, we have added the following sentence to the end of the limitations section: Future research should address how and why anthropometric characteristics and body composition might influence CrossFit performance.

Reviewer 2 Report

Menargues Ramírez_Body composition_IJERPH_2022 

I thank the authors for their efforts to improve the manuscript and to perform almost all the corrections proposed. I have only one more minor correction. 

Line 345: “An AI Index under 70% in men and around 70% in women seems to be 345 a desirable characteristic for CrossFit® athletes, although there is a lack of evidence about 346 the influence of AI index on performance.” Please, change AI to acromio-iliac index. 

Author Response

We appreciate the reviewer's comments and have made the suggested modification.